# Morbi-Mortality of the Victims of Internal Conflict and Poor Population in the Risaralda Province, Colombia

**DOI:** 10.3390/ijerph16091644

**Published:** 2019-05-11

**Authors:** Rafael Rentería-Ramos, Rafael Hurtado-Heredia, B Piedad Urdinola

**Affiliations:** 1Departments of Physics and Statistics, Universidad Nacional de Colombia, Cra 45 Bogotá, Colombia; 2School of Basic Sciences, Technologies and Engineering, Universidad Nacional Abierta y a Distancia de Colombia, 111321 Bogotá, Colombia; 3Department of Physics, Universidad Nacional de Colombia, Cra 45 Bogotá, Colombia; 4Department of Statistics, Universidad Nacional de Colombia, Cra 45 Bogotá, Colombia; bpurdinolac@unal.edu.co

**Keywords:** morbidity, mortality, mental health, social network, cardiovascular disease

## Abstract

This work studies the health status of two populations similar in most social and environmental interactions but one: the individuals from one population are victims of an internal armed conflict. Both populations are located in the Risaralda province, Colombia and the data for this study results from a combination of administrative records from the health system, between 2011 and 2016. We implemented a methodology based on graph theory that defines the system as a set of heterogeneous social actors, including individuals as well as organizations, embedded in a biological environment. The model of analysis uses the diagnoses in medical records to detect morbidity and mortality patterns for each individual (ego-networks), and assumes that these patterns contain relevant information about the effects of the actions of social actors, in a given environment, on the status of health. The analysis of the diagnoses and causes of specific mortality, following the Social Network Analysis framework, shows similar morbidity and mortality rates for both populations. However, the diagnoses’ patterns show that victims portray broader interactions between diagnoses, including mental and behavioral disorders, due to the hardships of this population.

## 1. Introduction

Victims of war and internal armed conflicts are some of the most vulnerable populations in the world since, due to the disruption provoked by those harsh conditions, they are exposed to a lack of basic public and health services, a weakening of social and family networks, and a loss of assets. When victims are forced to flee, all these conditions occur simultaneously. Indeed, refugees and internally displaced populations (IDPs) have much higher morbidity and mortality rates than the rest of the population for all observed conflicts in the world, as has been recorded from their experiences in refugee and IDP settlements or emergency assessments [1,2], as well as from refugees in their permanent places of refuge, which has been studied for Sweden and Denmark [3,4]. Violence typified in any of its forms, which is to say (1) intra state conflicts, typically referred to as wars, and (2) conflicts within the state, termed internal armed conflicts, have direct effects on mortality [5,6], on mental health, while increasing the vulnerability to sexual violence [7,8] of victims, as well as the indirect effects, which are measured by higher infant and maternal mortality rates [9,10,11]. The disruption that war and internal armed conflicts impose leads to extreme inequality in socioeconomic conditions between victims and the rest of the population, which in turn might be the underlying cause of those large differentials in morbidity and mortality, since this affects healthy lifestyles and all the other determinants of health [12].

This study aims to show the differences between the morbidity and mortality patterns of Colombian victims of internal conflict and those of a comparable subpopulation, namely an economically depressed population targeted by government subsidy programs in Colombia, in order to present a valid contrast instead of making a comparison with the rest of the population that holds a different socioeconomic status, and could thus up-bias the results. The findings are useful for a reconstruction process that aims to offer reparation to victims of conflict, such as the current process that has just started in Colombia, and for policy designs that intend to improve their quality of life. This study uses a novel and massive dataset for the Risaralda province in Colombia that consists of all the health records of each individual, with a 95% coverage of the total population in Colombia and, more importantly, it includes the self-reports of victims of the internal armed conflict. However, the data do not enable the discrimination of different types of human rights violations for those who self-identify as victims. The current statistics show that 80% of all the victims registered in official Colombian records are IDPs. Hence, this study presents the results for all victims of the internal armed conflict in Risaralda, knowing that in this province the vast majority are IDPs [13]. Therefore, it is most likely that the IDPs are leading the results presented here (We follow the international humanitarian law definition of a victim of an armed conflict. That is, any person who experiences a violation of at least one of their human rights from the international declaration of human rights (UN, 1948). In particular, any individual who has directly or indirectly suffered any kind of aggression or violence during an armed conflict as declared by the Red Cross International Committee in [7,8,9,10].)

Risaralda is one of the highest ranked provinces in Colombia in terms of socioeconomic conditions; it holds an almost universal coverage of public services (roads, water, electricity, gas, and sewage), and health services access, but it is located amid regions that have suffered persistent and intense armed conflict, attracting victims fleeing from zones with high violence, such as Chocó, Urabá, and the southwest provinces of Colombia. The living standards of victims of the internal armed conflict are so extreme that one expects worse physical and psychological conditions than those of the poorest population, who are already worse off than the rest of the population.

This study adds to the increasing literature on the application of network analysis and the relation between health and lifestyle disruption in three ways. First, we portray the different patterns of victims of violence and other subpopulations in an extreme case of endemic conflict, such as the Colombian conflict, where most victims of the conflict are not living in settlements or safe havens, and where the conflict has dramatically increased the number of refugees in other latitudes.

Second, we take advantage of the new possibility of cross referencing individual administrative records, which allows us to combine information on access to the health system, individual medical records, vital statistics, and the status of victims of the internal armed conflict in Colombia, a hard-to-reach subpopulation that is difficult or impossible to track with traditional surveys. We successfully cross referenced three official databases, all linked by the identification number of individuals, between 2011 and 2016: Base de Datos Única de Afiliados (BDUA, Unique Database of Affiliated (to the health system)), Registros Individuales de Prestación de Servicios de Salud (RIPS, Individual Records of Health Services), and Registro Unificado de Afiliación (RUAF, Unified Record of Affiliation). However, as with all big data, this novel dataset comes with challenges, and network analysis allowed us to identify the patterns of morbidity and mortality for both victims of internal conflict and the proposed control group (i.e., target populations for subsidy programs). The differences between these subpopulations lead us to identify the exact causes of death and predominant diseases for victims, namely cardiovascular and metabolic diseases. These findings are relevant for any policy design that intends to offer reparation to victims and improve victims’ living standards, since it enables the targeting of the exact diseases and prevention programs that are relevant for this particular population.

Third, an abrupt disruption in lifestyle leads to different observed health outcomes, and victims of the internal armed conflict are a perfect example of disruption, whether they have been internally displaced or not. A victim, per definition, suffers from a disruption of his/her normal habitat and lifestyle because the conflict has taken over his/her place of residency or has forced him/her to migrate to places without social networks or social capital. As a result, targeting programs to reestablish victims’ health should take these factors into account, and they should note that not only subsidized health treatment is needed, but that a combination of nutritional and psychological factors must also be incorporated into relief programs for the victims.

We are aware that using official records for the victim count of the internal armed conflict has several limitations. To begin with, the individuals registered in this dataset correspond to those that self-identified as victims of the Colombian internal armed conflict and were looking for governmental support programs of any kind, which imposes a bias from the total population of victims. However, the results could be considered as a lower bound of a larger issue. That is, we encourage readers to think of the population included here as the most vulnerable of all victims of the internal armed conflict, to the extent that, despite putting their anonymity in danger, they still looked for social welfare help from the state in order to survive. The total number of victims is higher, yet there might also be a higher heterogeneity in their socioeconomic conditions, even though other sources show that they are below the national average [13,14].

The remainder of this paper is organized as follows. This section describes the motivation. Section 3 describes the data and methods used, the next section presents the results, followed by the discussion, and the final section contains the conclusions.

### Motivation

Armed conflicts and war-like conditions have several negative effects on human health and development, as presented in recent literature (for a review, see [15,16]). Among all the populations affected by these conflicts, IDPs and refugees report the worst morbidity and mortality rates. Byarugaba [17] demonstrated low health results, both physical and psychological, for children and their mothers in IDP settlements in South Africa, which was driven by little or no access to health services. For instance, most of the causes of death for children under five years are preventable, including nutritional deficiencies, gastroenteritis, and dehydration. The World Health Organization [18] highlights three examples of increased infant mortality during conflicts due to communicable diseases: (1) measles, tetanus, and diphtheria became epidemics in Uganda during the mid-1980s, producing infant mortality rates twice as high as the total national rates; (2) in Zepa, perinatal and childhood mortality rates doubled during the Bosnia and Herzegovina conflict; and (3) in Sarajevo, the average birthweight fell by 20% in 1993 as a consequence of the doubling rate of premature births.

It is undeniable that the disruption created by war or internal armed conflicts affects socioeconomic conditions [15,19,20], which in turn change the lifestyles of all people, and this could translate into worse overall health conditions and mortality rates. However, in a long and persistent conflict, such as in the Colombian case, some skepticism remains about the lower life quality of the victims of the internal armed conflict [21]. On the one hand, the conflict has persisted for over five decades, with a shift in the aims of and the number of parties involved. On the other hand, the center of the conflict moved around the country, scattering victims across the entire territory. Both conditions make it very difficult to measure the victims of the conflict, and central official records are a novelty since there was no possibility of post-conflict or reconstruction terms while the conflict was ongoing. Additionally, the official statistical records of the entire population, censuses, and vital records in Colombia suffer from medium to high under-registration issues [22], which makes accountability a bigger issue.

Indeed, the measurement of victims of ongoing conflicts is very challenging. Brunborg and Urdal [6] highlighted the measurement complications of any conflict-related variable, ranging from the disruption of data collection by official and private sources, to interested parties involved in the conflict and struggling for power trying to show or hide certain numbers. For instance, in the Colombian context, the 2005 Colombian Population Census intended to directly measure the number of IDPs. Under the question “reason for migrating” asked to recent migrants, that is those who had moved within the previous five years, in addition to the traditional categories (job, education, health, and other) the possibility “due to violence” was added. The results could not have been more disappointing; they produced the lowest number of internally displaced populations among all available sources at the time, which included official registers, data from Non-Governmental Organizations (NGOs), and data from the Catholic Church. The reason was simple: in an ongoing conflict, any direct question will lead to under-registration because the victims of conflict feel the need for anonymity for their own protection.

To place the reader within the relevant context, this paragraph explains the dynamics of the Colombian internal armed conflict. The Colombian conflict erupted in the 1940s with the confrontation of followers from the two main political parties, the Conservatives and Liberals, in an era named “La Violencia” (The Violence), which systematically turned against unarmed civilians. By the 1960s, this conflict had evolved into a social class struggle with the eruption of communist guerrillas, as happened in many Latin American countries. By the 1980s, paramilitary groups arose as a force against leftist guerillas, and by the 1990s several armed groups found profitable violent ways to achieve political and economic aims, including the systematic displacement of populations to steal land, drug production and trafficking, weapons trafficking, kidnapping, and extortion. Indeed, the most violent decade in Colombian history during the past century was the 1990s [16]. By the turn of the century, the president in office had counteracted a full military attack by guerilla groups and led a demobilization process of paramilitary groups. Afterwards, in November 2016, the next administration signed a peace agreement between the Colombian government and the largest guerrilla group, FARC (Fuerzas Armadas Revolucionarias de Colombia). By 2018 a negotiation process between the government and the next largest and oldest guerrilla group, ELN (Ejército de Liberación Nacional) had begun; however, by February 2019, those dialogues broke down due to a terrorist attack on the Police Academy in Bogota. The agreement began to be implemented in 2017, with the hope of ending the armed internal conflict [9].

One of the largest demographic effects of this persistent and intense conflict is that Colombia has the second largest IDP in the world, with over six million people, many of whom were also forced into dispossession [18]. When IDPs arrive at their destination, they live in poverty, despite the social welfare programs that are implemented [23], since those programs are insufficient to provide for all the basic needs of forced migrants, and no health prevention program targets this particular population and their needs. Indeed, over the past two decades in Colombia, the IDP has shown much lower socioeconomic, housing, and employment conditions than before the displacement [24,25], as evidenced by lower sanitary conditions and less access to health facilities and services, which is reflected in worse health outcomes. However, previous studies have not linked the health outcomes of an IDP with their lifestyles, as we propose to in this study, even though this might be more beneficial for identifying the particularities of this population, as a means of enabling the design of a special prevention program that could be more cost-effective and therefore more successful in improving their health outcomes.

The empirical results of similar studies in other contexts support the hypothesis we propose. Audet [26] showed the negative effects of habits or lifestyles on vulnerable populations, particularly on women, linked to obesity and chronic diseases. Brown [27] used the social networks analysis approach to associate socioeconomic and environmental variables with nutritional lifestyles in Los Angeles; they studied the restriction imposed on fast food and its implications on illnesses related to obesity, and found that unhealthy inexpensive food and poverty were structurally related.

## 2. Materials and Methods

We all live in a biological and social network, and the specific pattern of interactions of a person with his social and environmental surroundings determines many aspects of his state of health. Everywhere, the morbidity and mortality outcomes of individual patients provide information about health determinants and risk factors for individuals and groups. Here we construct Ego networks of morbidity for each patient, using his medical records, and superpose those of a specific population. Through the topological properties of the networks it is possible to identify differences between some populations; in this work we compare two populations similar in most social and environmental interactions.

We linked the records obtained for the Risaralda province from three independent administrative databases for all the years between 2011 and 2016, by using a unique identification number. The datasets are available for research purposes, and they are collected and managed by the Colombian Ministry of Health. All the data is anonymized, and the identification number is unique among all the datasets, which enables the cross referencing and matching of records, while revealing no personal information according to the data privacy laws in Colombia (Publications using these datasets cannot publish individual information but can publish aggregated information, in accordance with personal information law in Colombia (Ley 1266 de 2008), and we have followed this in this article).

We pooled all of the information from 2011 to 2016 for two reasons. First, the Colombian State only started to implement a system to amend victims of the internal armed conflict in 2011, according to Law 1448 of the year 2011 [28], which clearly states that benefits and care measures should be provided to all self-reported victims of the conflict since the year 1985. This law defines, as we have described before, a victim as any person who has suffered a violation of his/her fundamental human rights or the international human rights due to the events occurring during the internal armed conflict, and this provided the framework for this research focusing on the data at hand. Second, since the data for a single year might record an exaggerated mortality increment or decrement for reasons unrelated to the conflict, for example a mortality shock due to a new epidemic, a natural disaster, or a lack of resources devoted to the vital registration system, we instead used the six years pooled together.

The first dataset that was used was the Base de Datos Única de Afiliados (BDUA, Unique Database of the Affiliated (to the health system)), which contains individual records of the affiliated users of the health system in Colombia. There are two possible ways to become part of this dataset: by making payments to the system (the contributive regime) if the person is employed or self-employed in a formal job, or by being subsidized by the government (the subsidized regime) if the person is unemployed or working under very poor conditions. Dependents are also added to the database, that is children under the majority age, or below the age of 25 if they are full time students, as well as the elderly and spouses. Altogether, 96.6% of the population in Risaralda belongs to either category, and therefore is included in the dataset. Both the victims of internal armed conflict and populations targeted for health subsidies belong to the subsidized regime; 52.2% of the population is in the contributive regime and 44.4% in the subsidized regime. The targeted populations include those who are unemployed, with no labor income, and with no direct family member who could pay any contribution to the health system for them.

The victims of internal armed conflict are recognized in BDUA as a distinct category of self-reported vulnerable populations once they register in the system or update their information. From a database management point of view, the victim category is the only one that remains permanent in time. All other categories add to this condition, whereas for all other categories the mark up changes. For example, a person who reports as an indigent and later changes her status to indigenous population enters the dataset in this last category. Instead, a person who first reports as a victim and later as an indigent will hold both conditions in the dataset. Likewise, if the person is part of the welfare system that targets the poorest of the poor in Colombia, he/she is part of the Identification System of Potential Beneficiaries for Welfare Programs (Sistema de Identificación de Potenciales Beneficiarios de Programas Sociales, SISBEN). That information is collected by surveying the population in the focus areas across Colombia, through the National Planning Department, and anyone who believes they should be included in the program but was not surveyed or suddenly fell into poverty can demand a survey from his/her local major. This survey provides a multidimensional poverty index that ranks households into six poverty categories, with the poorest of the poor ranked in the system as SISBEN I and SISBEN II. The index construction follows a secret formula to avoid moral dilemmas. Therefore, anyone ranked as SISBEN I or II immediately becomes part of the subsidized health service and marked in the BDUA dataset as such. This affiliation database is continuously updated, like most administrative health record systems in the world, since the population constantly changes its relevant status for the system, for instance changes in employment status, changes of jobs, and changes of residency. BDUA contains other relevant demographic information for this work, such as age, sex, and ethnicity.

The second dataset, Registros Individuales de Prestación de Servicios de Salud (RIPS, Individual Records of Health Services), holds the full medical records of all patients who are part of BDUA and includes information of who attended regular medical appointments or received urgent care, and the follow-up of medical exams and treatments. The information contains symptoms, diagnoses, and prognoses, as recorded by medical doctors. This information is essential for the morbidity profiles. For the mortality diagnoses, we used the main diagnostic, and all other three underlying causes were dismissed for simplicity.

Finally, Registro Unificado de Afiliación (RUAF, Unified Record of Affiliation), contains vital records, including birth and death certificates, that aim to be universal. One of the advantages of the vital registration system in Colombia is that death records include the main cause of death as well as the underlying causes of death, recorded under the International Diseases Classification in its 10th version (ICD-X). For this study, we only used the main cause of death, and all other underlying causes were reported in less than 5% of cases and were therefore not used.

The dataset built from the BDUA, RIPS, and RUAF databases contains 970,000 people (96.7% of the inhabitants of the Risaralda province), with 48,422,050 confirmed diagnoses, 242,211,050 procedures (used as a metric to eliminate disease suspicions), and 23,648 mortality records for the specific cause of the diagnoses. This information was obtained from 9,684,410 medical visits (general, emergency, and follow-up) of the inhabitants of Risaralda who used the medical system between the years 2011 and 2016. Just as a reminder, 2011 was the first year that the Colombian government implemented the victims’ reparation law, which among other benefits includes the subsidized access to the health system.

Table 1 summarizes cause-specific morbidity and mortality rates for the entire period under study for the most common diseases in the Risaralda province, and summarizes why using traditional basic rates does not allow a deep characterization of the health issues of victims of the armed conflict, using these large administrative records.

The morbidity and mortality rates portrayed a similar pattern across the categories, with a very much expected trend in most cases that was higher for vulnerable populations than for the rest of the population. Panel A includes crude morbidity rates and portrays very similar rates for all causes for the three categories presented here: victims of the internal armed conflict, the SISBEN I–II population, and the rest of the population in Risaralda (Others), except for the rate of mental and behavioral issues, which for female victims was four times higher than for SISBEN I–II and seven times higher than for the rest of the population, and which, for males, was more than twice that of SISBEN I–II and almost twice that of the remaining population.

Similarly, the mortality rates followed the same pattern, with similar rates across the causes of death for all three categories, except for mental and behavioral disorders, which monotonically increased from others to SISBEN I–II, while victims held the highest mortality rate. For all other causes, the rates were fairly similar, despite some small differences in mortality and morbidity; for example, metabolic diseases were more prevalent among females in SISBEN I–II than among victims or the rest of the population.

It is precisely the potential of individual records and the fact that traditional rates could mislead health outcomes for minorities and vulnerable populations that led this study to use the social network analysis approach. Following the definitions in [29,30,31,32], a network represents a system through its constituents and the relationship between them. The network can be mapped onto a graph G(V,E), where V is the set of vertices v∈V, and E is the set of edges or links between the vertices. This methodology has been widely used to investigate several natural, social, and artificial systems, including health issues such as vector spread diseases [33,34], the link between mobility and he the spread of diseases [35,36], the spread of epidemics [37], and lifestyles and health [38]. Studies linking mobility and epidemics are relevant for the aim of this study, since victims fleeing an armed conflict, mostly IDPs, could introduce diseases to populations in the arrival places, which would never have occurred if the conflict had not existed [25,26]. Furthermore, their low living standards and their associated lifestyles, such as a low nutritional intake, limited access to healthcare and services, exclusion from the public health system, and low sanitary conditions, lead to a particular morbidity profile [9,10,16,18].

### 2.1. Morbidity Network of Diagnoses

**We began with the aggregate information of the social network, namely all the diagnoses in the Risaralda province, for both populations under study**. The population that qualifies for subsidy programs in Colombia is ranked as SISBEN and classified at the levels I and II. We defined the diagnoses (ICD-X) as the nodes of the network. The edges were defined as the coupling between two diagnoses, given by their co-occurrence in each medical event, for every individual in the RIPS (ego-networks), including the main and up to three secondary medical diagnoses, in all kinds of medical events: regular consultations, emergency consultations, treatments, and follow-ups, as shown in Figure 1.

The nodes or vertices represent the diagnoses (International Diseases Classification in its 10th version, ICD-X), $D_{1}$ is the main diagnosis, while $D_{2}$ and $D_{3}$ are secondary diagnoses. The edges or links represent the co-occurrence of primary and secondary diagnoses for every individual in each medical event: regular consultations, emergency consultations, treatments, and follow-ups (Source: authors’ scheme).

In Figure 1, $D_{1}$ is the main diagnosis, while $D_{2}$ and $D_{3}$ are the secondary diagnoses. The weight of the links $W_{ij}$ is defined as:(1)$$W_{ij}=\underset{k}{\sum }D_{ik}.{\hat{D}}_{kj},$$
where the elements of the matrix D, $D_{ik}=1$ ifdiagnosis $D_{i}$ was recorded at RIPS in the medical event k, and $D_{ik}=0$ otherwise. $\hat{D}$ is the transposed matrix of D.

The graph of the network obtained by applying the methodology described above is shown in Figure 2.

The nodes are diagnoses (International Diseases Classification in its 10th version, ICD-X), and the links correspond to pairs of diagnoses present in medical records from all kinds of medical events: regular consultations, emergency consultations, treatments, and follow-ups. 

The degree of node i, $K_{i}$, is the number of adjacent nodes to i; the strength of node i is defined as $F_{i}=\sum_{j}W_{ij}$. The diagnosis I10X, which is essential (primary) hypertension, had the largest degree and strength: 1259 connections for 9918 medical events, followed by urinary tract infection (N390), acute nasopharyngitis (common cold) (J00X), gastritis unspecified (K297), hypothyroidism unspecified (E039), intestinal parasitism unspecified (B829), and low back pain (M545).

The degree and strength probability distributions of the network, p(k) and p(f), respectively, can be represented using the complementary cumulative distribution function, defined in Equation (2), which produces Figure 3:(2)$$P_{K}=\int_{k}^{\infty }p\left(k^{\prime }\right)\;dk^{\prime };\;F_{K}=\int_{k}^{\infty }p\left(f^{\prime }\right)\;df^{\prime }$$

Figure 3 shows the existence of three regions. First, for $P_{K}$ in the interval [$10^{0}$,$10^{-1}$], several nodes had few connections and small weight. Second, between $10^{-1}$ and $10^{-2}$, the nodes were strengthened. Third, for $P_{K}<10^{-2}$, few nodes were highly connected in the fat tails, neither in degree nor in strength.

We cut the network by selecting the nodes with a strength of $F_{k}>10^{1}$. This allowed us to center the analysis on interdependent and correlated diseases that sketch morbidity paths, mostly related to prevalent conditions. This network reduction provides a better understanding of the morbidity patterns in Risaralda for both subpopulations of interest. To detect morbidity patterns, we used cohesion properties, specifically cliques of maximally complete subgraphs, as described in the following subsection.

### 2.2. Selection Algorithm for Detecting Communities

For this study, we needed to fully identify morbidity patterns for both subpopulations of interest, namely the victims of internal conflict and beneficiaries of the subsidy programs SISBEN I and II. To do so, we chose the k-communities algorithm of [39,40], since it keeps the superposition of diagnoses in the subgraphs, even though many individuals share the same initial diagnosis, but with different final diagnoses.

For this algorithm, we needed to first quantify the total number of cliques in the network following [41] (see Figure 4a). In this case, the maximum number of diagnoses per medical event was k=4, and this becomes the value k for percolation. The second step was to find the adjacent cliques, which are those that share k−1 nodes. Again, we followed [39,40] (see Figure 4b,c). The final step was to join all of the cliques reached by the series of adjacent cliques to conform the *k*-community (see Figure 4d). Figure 4 shows each step of this algorithm, which enabled us to distinguish between the two subpopulations of interest. For consistency purposes, it was necessary to analyze each subpopulation’s subgraphs, for which we used a motif analysis.

### 2.3. Intensity Analysis and Motif Coherence

In social networks analyses, motifs are interconnected patterns with a much larger frequency than random subgraphs [42]. They are common in systems studied in biology [43,44] and ecology [45,46], among others. Motifs have intrinsic characteristics that condition the probability of occurrence of certain values of degree, despite their application in making particular cuts of the network [42]. This enables the capture of a series of trends in the network circumvent information, such as the nodes’ consensus, that control their flows. This characteristic is essential to associate diagnoses and illness with lifestyles.

To incorporate topological aspects of motifs in weighted networks (and the strength), we used the intensity metrics and motif coherence developed by [47]. Intensity I(g) for subgraph g with vertices $V_{g}$ and edges $l_{g}$ is defined as:(3)I(g)=(∏(kj)∈lgWkj)1|lg|

This guarantees a link qualification in motifs from the $W_{kj}$ values and led to prioritizing communities of diagnoses that build up a morbidity landscape. After that, we established the coherence as Q(g), which enabled us to study the consensus between the nodes from the edges in the motifs. Coherence takes the values close to the most important unit in its subgraph to establish an association between subpopulations, and it is defined as the ratio between the intensity I(g), and the geometric mean of their weights, $W_{kj}$, as presented in Equation (4):(4)$$Q\left(g\right)=\frac{I_{g}}{\sum_{\left(ij\right)\in l_{g}}^{n}W_{kj}}$$

### 2.4. Mortality Network Analysis

Mortality is a much simpler network than morbidity and does not require all of the morbidity data, as mentioned in Section 2.1, Section 2.2 and Section 2.3. The structure of this network is described in Figure 5. Node C1 was the main cause of death in the diagnosis, and C2 and C3 were the secondary or underlying causes of death.

## 3. Results

First, we present the morbidity and mortality networks for the victims of internal armed conflict, as recorded in the BDUA dataset, before presenting the control group or those who classify as the poorest of the poor in Risaralda, categorized as SISBEN I–II.

### 3.1. Results for the Victims of Internal Armed Conflict

Figure 6 is the complementary cumulative distribution function that resulted from applying the methodology described below. It summarizes the networks of diagnoses for all of the individuals included in the BDUA database that were identified as the victims of armed conflict.

The results were similar to those of the morbidity network for the whole Risaralda province, in the full range of $P_{K}$ for $K_{i}$ and $F_{ki}$. The algorithm described in the methods section enabled the intensity and coherence of the motifs to be measured and for communities to be detected. Figure 7 presents the results, showing the coherence values for the two motifs selected to analyze the network of diagnoses for the victims. In both cases, the motifs had low coherence values, between 0.05 and 0.1, with the exception of a small set with values close to 0.3. The motifs with the highest coherence were the cliques labeled 80 and 82, as well as community 50, composed by the diagnoses in Figure 8.

The mortality network results are summarized in Figure 9. Unsurprisingly, there was some overlapping of the three denoted diseases. Both diabetes mellitus without complications, non-insulin dependent (E119), and hypertensive heart disease without (congestive) heart failure (I119) were common to both results, denoting the final fatal outcome of both illnesses. The remaining disease in Figure 9 is diabetes mellitus with multiple complications, insulin dependent, which is mostly due to genetic conditions; however, some type II diabetic patients might develop type I diabetes mellitus if the risks of the former are not controlled, despite medical treatment.

### 3.2. Results for the Population Classified as SISBEN I and II

The results are shown in Figure 10 and Figure 11; for simplicity, we refer to this population as SISBEN I and II. It is important to remind the reader that we chose this population as a contrast to victims of the internal armed conflict, since both suffer high socioeconomic burdens, and SISBEN I and II are a better control than the remaining population. The pattern in the degree distribution and node strength for morbidity was very similar to that reported for the victims of internal armed conflict. Therefore, the cut was made at the same interval as before, and the results are portrayed in Figure 11.

By contrast, the motif coherence levels for SISBEN I and II were dissimilar to those of the victims of the armed conflict, despite most cliques being between 0.05 and 0.1. In fact, the highest coherence level was 0.27 and allowed the construction of Figure 12. Despite one clique having this 0.27 level, it overlapped with its own value and was replicated for the entire network, showing that it had the largest influence for most diseases among SISBEN I and II.

This motif was smaller than that of the victims of armed conflict; however, it shared one diagnosis, diabetes mellitus without complications (E119). The other two diseases, pure hyperglyceridaemia (E781) and primary hypertension (I10X), were not fatal outcomes for victims of internal armed conflict, but the former was part of the morbidity network. In general, these results show differences between the populations. Nonetheless, unhealthy lifestyles and cardiovascular diseases led to similar but different mortality outcomes. For instance, a hazard risk for hypertension, beyond an unhealthy lifestyle, is diabetes mellitus [48,49,50,51]. Thus, the causes of these fatal consequences are all connected and, in most cases, linked to unhealthy lifestyles, as previously described.

## 4. Discussion

Both populations under study are, without question, highly vulnerable to poor health conditions, due to the extreme socioeconomic conditions they face. This study’s results showed an unhealthy lifestyle as the common ground for the different diseases. More importantly, the differences in the final causes of death reported in each case, which result in access to health treatments and prognoses for both populations, were different.

The similarities suggest that diabetes mellitus without complications (E119) and pure hyperglyceridaemia (E781) could lead to high blood pressure and high blood cholesterol levels, both ending in fatal consequences related to cardio vascular diseases. According to [52,53,54,55,56], the source of these diseases is related to the lack of a healthy lifestyle, which includes diets based on a high consumption of saturated fat and/or carbohydrates, including processed sugar, a lack of physical activity, and a frequent consumption of tobacco and alcohol. This unhealthy lifestyle is also linked to cerebrovascular diseases, including hypertension and metabolic diseases [57]. We believe these risk factors could be higher for victims of the internal armed conflict than for SISBEN I-II populations due to results from other studies [58], but this needs further study.

The main difference between the two populations was that the victims of armed conflict suffer from mixed anxiety and depressive disorder (F412), as well as dysthymia (F341). This is not a surprising result, since victims of conflicts, war, or internal armed conflict have a much higher incidence of mental health issues [58,59,60]. However, a surprising result was the pattern that linked mental and behavioral disorders (F341 and F412) to the outcomes related to cardiovascular and metabolic diseases (E781, E039, E782, I119, and E119), as shown in Figure 2.

In the literature, some have linked post-traumatic stress with these metabolic diseases [59,60,61], but causality has not been established with hypertension, cardiac failure, coronary heart disease or metabolic diseases [62,63,64]. Recent literature connects mental and behavioral disorders with the proper functioning of the endocrine system. The metabolic system depends directly on the wake and sleep hours of the body (circadian cycle), and the sleep quality directly links to glucose regulation and hormonal abnormalities, which can affect sleep quality, suggesting a double causal relationship between both of them (for a medical literature review on this result, see [65]). Moreover, genetic studies are finding a particular link between both: “GWA studies have linked *PER3* to T2DM (256) and *CRY2* with fasting glycemia and insulin concentrations (257). In these studies, melatonin receptor 1B (*MTNR1B*) variants are also consistently associated with insulin secretion and T2DM risk (142–144).” [65].

The results for this very particular group of victims of the conflict in Risaralda seem to reinforce this medical knowledge, which is the key important result that is applicable to the design of a public policy that intends to improve the health quality of this population. Not only do these victims of the conflict need the prevention campaigns designed for the poorest populations, which encourage the switch from an unhealthy to healthy lifestyle, but also psychological and/or psychiatric help. In fact, we believe that this is at least as important as their basic physical health, since their condition reinforces diseases with a tragic outcome. For instance, mixed hyperlipidaemia (E782) and pure hyperlipidaemia (E781) are directly linked to hypertensive heart disease (I119).

## 5. Conclusions

Despite evidence in the literature showing that victims of war or armed conflicts have worse health conditions than the rest of the population, morbidity and its link to mortality for the victims of the internal armed conflict in Colombia had not been studied in depth. In part, this is due to the lack of reliable data. However, this paper presents enough evidence from newly released administrative records with health-related micro data to show the main morbi-mortality causes for the victims of armed conflict, as recorded in those databases.

Combining micro data from massive datasets, such the administrative records at hand, is very demanding, but linked unique identification numbers enables this process and produces a new data source worth exploring. More challenging, however, is the analysis of such large datasets. The proposed methodology, based on a social networks analysis, led us to fully identify the exact morbidity and mortality patterns from an ocean of chaotic information, and was the best option for these kinds of research questions and data.

To establish a reference point, we analyzed the morbi-mortality frequencies of the victims of internal armed conflict and the poorest population (SISBEN I and II), residing in the Risaralda province in Colombia. The results demonstrated that both subpopulations were vulnerable and suffered from non-communicable diseases mostly related to unhealthy lifestyles, such as type II diabetes, hypertension, and hyperglyceridaemia. However, the victims of armed conflict showed mortality outcomes that demonstrated a deepening of those conditions (e.g., type I diabetes). Indeed, victims of conflict showed a very interesting connection between metabolic diseases with cardiovascular mortality outcomes and the presence of mental and behavioral disorders. This finding is very much in line with recent literature that relates metabolic disorders and mental health, and is the key finding applicable to the design of a program that aims to increase the physical health of victims of the internal armed conflict in Colombia: victims need both mental and physical health support to overcome health issues.

## Figures and Tables

**Figure 1 ijerph-16-01644-f001:**
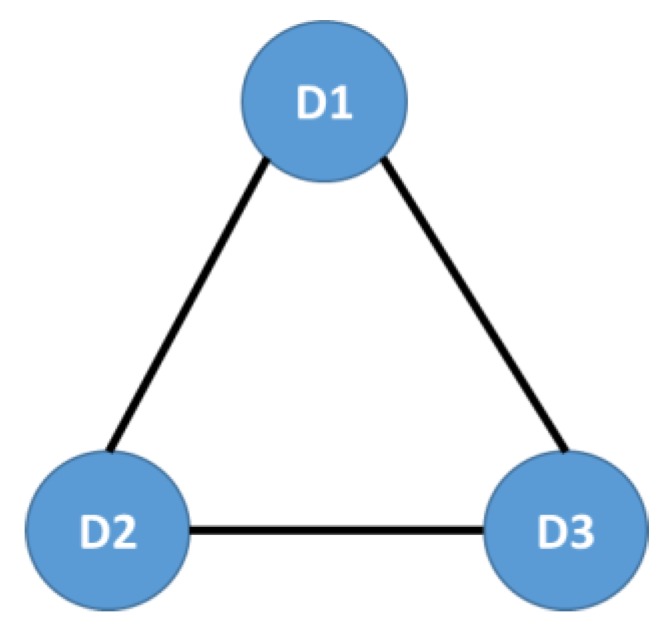
Structure of the Morbidity Network.

**Figure 2 ijerph-16-01644-f002:**
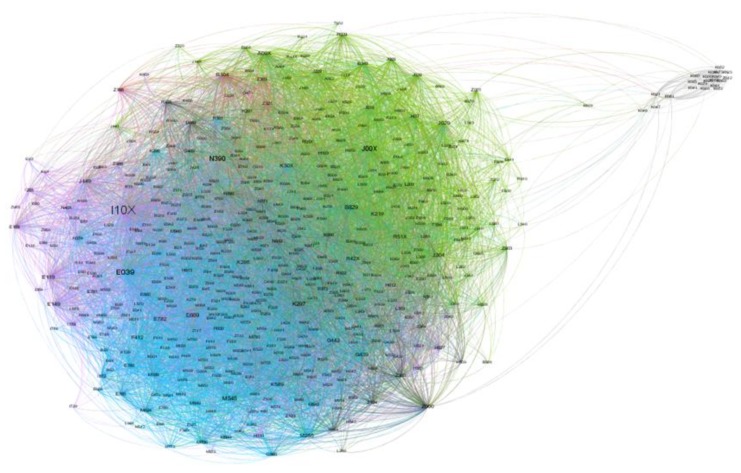
Graph of the Morbidity Network of Diagnoses for all medical records of residents in the Risaralda province, from 2011 to 2016. Source: authors’ own calculations.

**Figure 3 ijerph-16-01644-f003:**
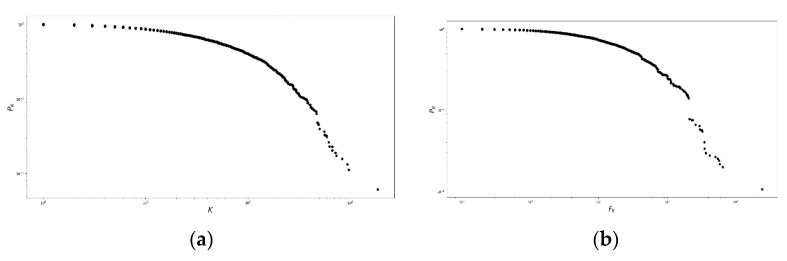
Nodal degree and nodal strength for the population residing in the Risaralda province. Complementary cumulative distribution functions for (**a**) the nodal degree probability distribution and (**b**) the nodal strength of the Morbidity Network of Diagnoses for the total population resident in the Risaralda province, from 2011 to 2016. Source: authors’ own calculations.

**Figure 4 ijerph-16-01644-f004:**
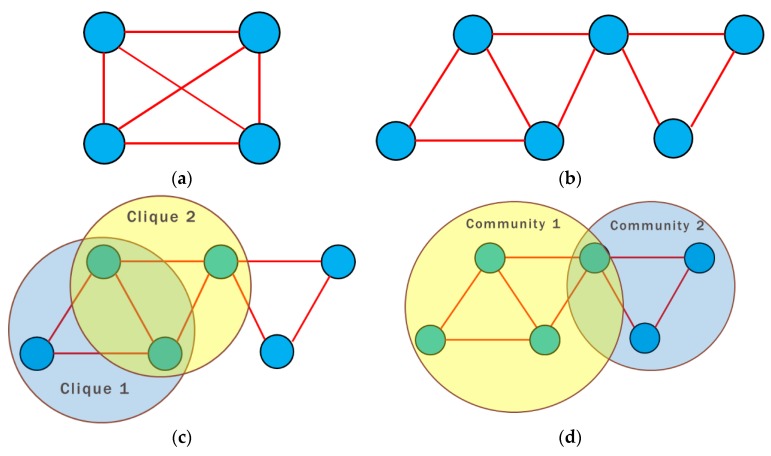
Detection algorithm for k-communities. (**a**) Cliques detection with size k (**b**) Detection of the sequence of Cliques (**c**) Selection of the Clique’s neighbors with size k-1 (**d**) Formation of K – Communities. Source: authors’ own calculations.

**Figure 5 ijerph-16-01644-f005:**
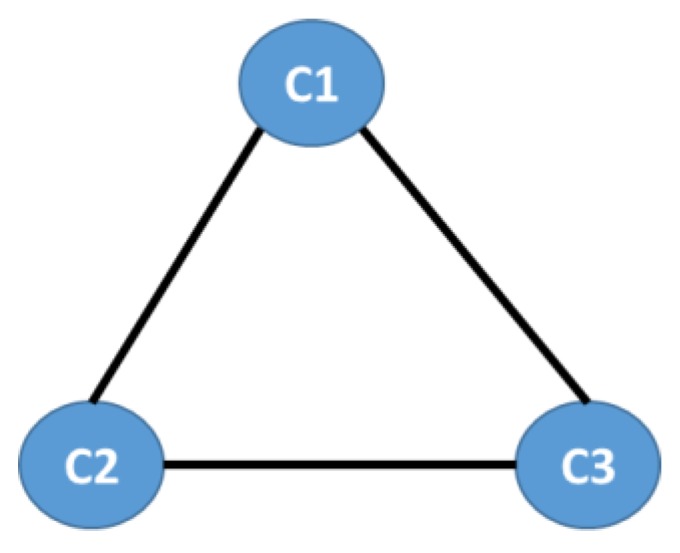
Structure of the Mortality Network. The nodes or vertices represent the causes of death: $C_{1}$ is the main cause, while $C_{2}$ and $C_{3}$ are the secondary causes. The edges or links represent the number of cooccurrences of both primary and secondary causes for every individual. Source: authors’ scheme.

**Figure 6 ijerph-16-01644-f006:**
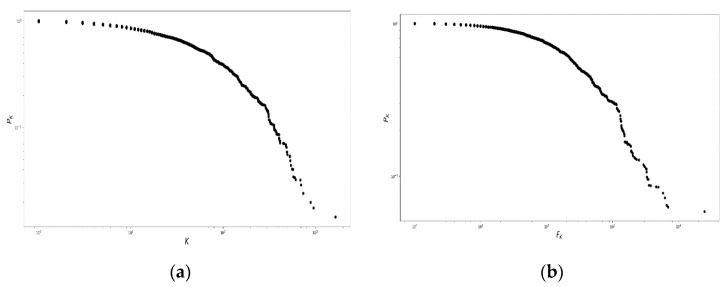
Nodal degree and nodal strength for the victims of the armed conflict in Colombia, resident in the Risaralda province. Complementary cumulative distribution functions of (**a**) the nodal degree *K*, and (**b**) the nodal strength $F_{K}$ for the Morbidity Network of Diagnoses for the victims of the armed conflict in Colombia, residing in the Risaralda province, from 2011 to 2016. Source: authors’ own calculations.

**Figure 7 ijerph-16-01644-f007:**
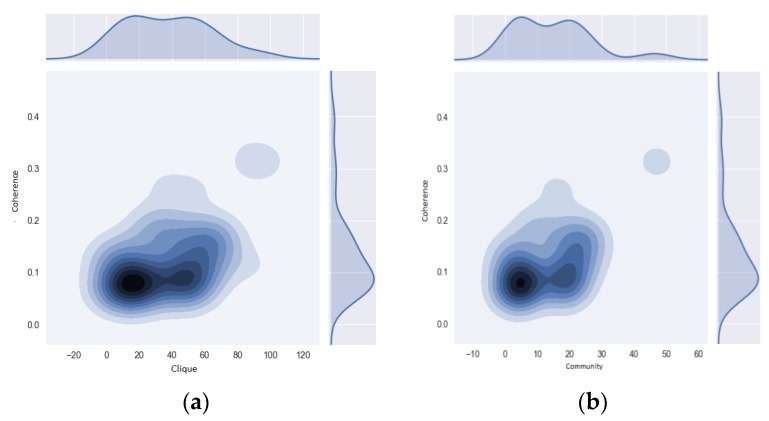
Motif coherence for cliques and communities. Motif coherence for (**a**) cliques and for (**b**) communities for the Morbidity Network of Diagnoses of the victims of the armed conflict in Colombia, residing in the Risaralda province, from 2011 to 2016. Source: authors’ own calculations.

**Figure 8 ijerph-16-01644-f008:**
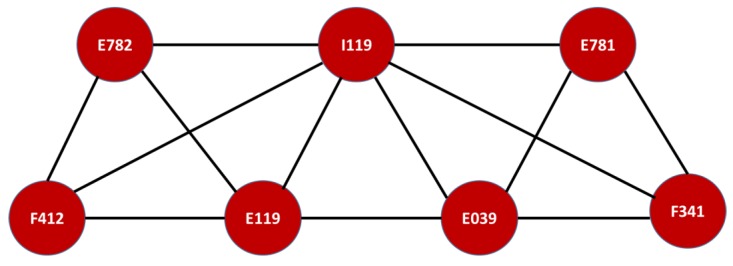
Motifs of diagnoses with the highest coherence level. Morbidity Network of Diagnoses for the victims of the armed conflict in Colombia, residing in the Risaralda province, from 2011 to 2016. Source: authors’ own calculations. Note that the ICD-X codes correspond to: E782: mixed hyperlipidaemia, I119: hypertensive heart disease without heart failure, E119: type 2 diabetes mellitus without complications, E039: hypothyroidism, unspecified, E781: pure hyperglyceridemia, F341: dysthymic disorder, and F412: mixed anxiety and depressive disorder.

**Figure 9 ijerph-16-01644-f009:**
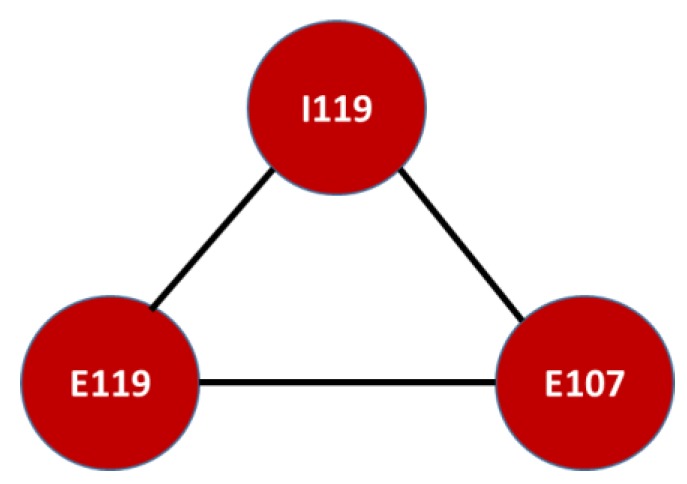
Clique with the highest overlap in morbidity and mortality networks. Mortality Network of Diseases for the victims of the armed conflict in Colombia, resident in the Risaralda province, from 2011 to 2016. Source: authors’ own calculations. Note that the ICD-X codes correspond to: E119: type 2 diabetes mellitus without complications, I119: hypertensive heart disease without heart failure, and E107: diabetes mellitus insulin dependent, with multiple complications.

**Figure 10 ijerph-16-01644-f010:**
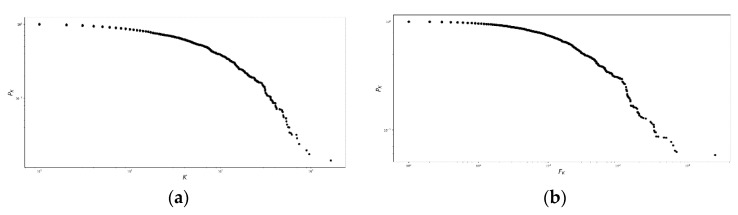
Nodal degree and nodal strength for the SISBEN I and II population resident in the Risaralda province. Complementary cumulative distribution functions of (**a**) the nodal degree *K*, and (**b**) the nodal strength, $F_{K}$ for the Morbidity Network of Diagnoses of the SISBEN I and II population, residing in the Risaralda province, from 2011 to 2016. Source: authors’ own calculations.

**Figure 11 ijerph-16-01644-f011:**
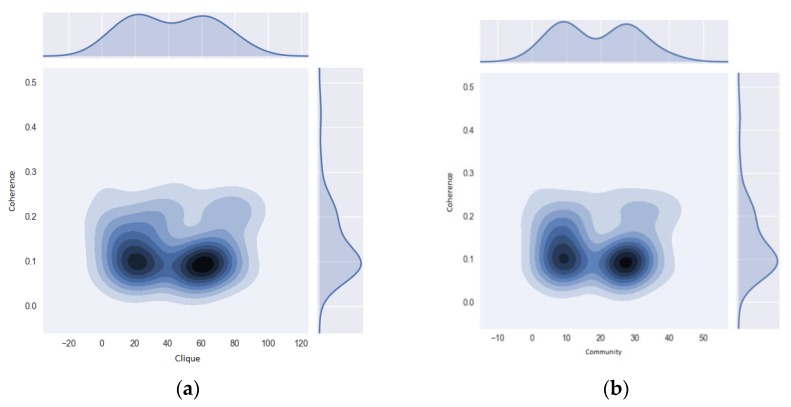
Motif coherence for cliques and SISBEN I and II. Motif coherence for (**a**) cliques and (**b**) SISBEN I and II for the Morbidity Network of Diseases for victims of the armed conflict in Colombia, residing in the Risaralda province, from 2011 to 2016. Source: authors’ own calculations.

**Figure 12 ijerph-16-01644-f012:**
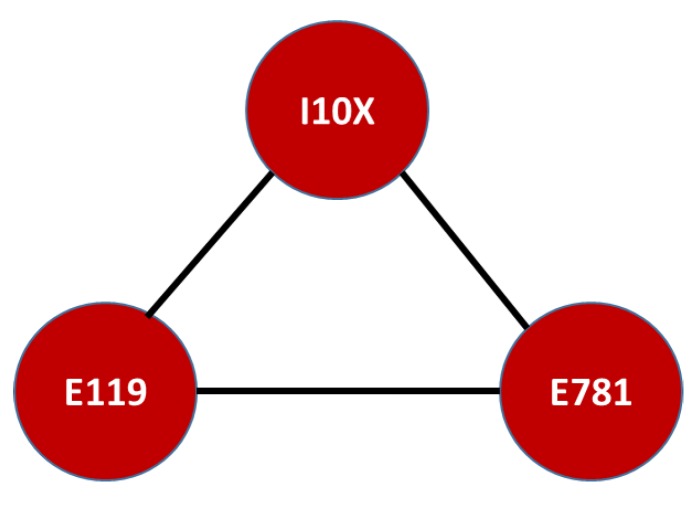
Highest overlapping cliques in morbidity and mortality for SISBEN I and II. Mortality Network of Diseases in the SISBEN I and SISBEN II population residing in the Risaralda province, from 2011 to 2016. Source: authors’ own calculations. Note that the ICD-X codes correspond to: E781: pure hyperglyceridemia, I10X: essential (primary) hypertension, and E119: type 2 diabetes mellitus without complications.

**Table 1 ijerph-16-01644-t001:** Morbidity and mortality rates in Risaralda, 2011–2016.

**Panel A. Cause specific morbidity rates per 100,000 inhabitants in Risaralda for victims, SISBEN I–II, and others**										
Population Group	Cardiovascular		Communicable Diseases		Mental and Behavioral		Metabolic		Nervous System	
	M	F	M	F	M	F	M	F	M	F
Victims	0.09	0.14	0.09	0.14	0.07	0.15	0.11	0.13	0.09	0.13
SISBEN I-II	0.07	0.15	0.10	0.12	0.09	0.04	0.14	0.18	0.10	0.11
Others	0.07	0.09	0.08	0.14	0.04	0.02	0.10	0.12	0.09	0.07
**Panel B. Cause specific mortality rates per 100,000 inhabitants in Risaralda for victims, SISBEN I–II, and others**										
Population Group	Cardiovascular		Communicable Diseases		Mental and Behavioral		Metabolic		Nervous System	
	M	F	M	F	M	F	M	F	M	F
Victims	0.05	0.08	0.04	0.08	0.07	0.05	0.10	0.10	0.07	0.10
SISBEN I-II	0.09	0.06	0.05	0.08	0.05	0.03	0.10	0.12	0.08	0.09
Others	0.07	0.14	0.05	0.06	0.02	0.01	0.10	0.14	0.06	0.07

F: Females, M: Males.; Source: authors’ calculations using RIPS and RUAF.

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
