# Peer review of "Morbi-Mortality of the Victims of Internal Conflict and Poor Population in the Risaralda Province, Colombia"

_ijerph, 2019, doi:10.3390/ijerph16091644_

Reviewer 1 Report

Thank you for the opportunity to review this manuscript. This is a great piece with very innovative methodological techniques. These are some comments to improve it.

1. The abstract must clearly state the puzzle or research gap the study attempts to fulfil. The authors understandably state the objective of the paper, but it is not clear why this needs to be researched.
2. In the first paragraph of the introduction, it would be great if the authors could beef up the background issues with some underlying statistical data on differences in mortality and morbidity refugees and IDPs and the rest of the populations. Examples could be used from places other than Colombia
3. Again, just as I mentioned in the background, please present the research gap that this paper seeks to address. Perhaps, as part of the second paragraph in the introduction. I see that this was attempted in the ‘motivation’ section. However, this must be presented early so anyone reading would be guided by it from the onset.
4. Materials and methods: please justify why you used the data from 2011-2016
5. While the state clearly states the use of network analysis, I think it will be much better to have the raw data on mortality and morbidity presented in tables. Presently, the figure (2) to demonstrate the incidence of morbidity is not legible.
6. I see very much limited attempt discussing the results. Perhaps this is because the authors combined the results with the discussion section. I strongly advise that the two sections are separated so that adequate attention could be given to what the results mean and its implications for health-related well-being of refugees and IDPs as well as public health.
7. In the conclusion section, the paper is missing the key take-home message. The section is just a rehash of the methodology and results. Please reflect on the most important contribution of the paper and present it at this point. You could take a cue from the research gap presented under ‘motivation’ to provide the contribution of the study. This message must also be indicated at the end of the abstract.

Reviewer 2 Report

Firstly, thank you for the opportunity to review this paper on this important topic; covering a topic (IDPs) which is under-researched. I attach comments for consideration and suggestions of changes below:
The title highlights victims of internal conflict and poverty - however, the introduction focuses on refugees and internal displacement.  There therefore seems to be some discrepancy: is this article primarily focusing on forced displacement as a cause of morbidity and mortality or conflict, violence or poverty directly as a cause of increased morbidity and mortality?  This needs to be made clearer throughout as the impact of displacement itself can be different to the direct effects of conflict or violence.  

From Line 54: Definitions and terminology need to be clarified throughout:

(a) it is unclear what the terminology 'victims' refers to here and in the subsequent paragraphs - is this victims of violence or those affected by conflict or those who are internally displaced from conflict?  Consistency in the use of international definitions correctly throughout and in the title will strengthen this article.

(b) Differentiation between conflict and violence: it is necessary from the beginning of the article to highlight the changes globally in types of conflict: from predominantly inter-state wars to violent conflict within states-  including an increase in protracted conflict and gang violence.  These conflicts disproportionately affect civilians.  The conflict in Columbia would therefore fall into the category of protracted internal conflict and gang violence.  This is also relevant for the literature review which in my opinion should have more focus on other contexts of protracted internal conflict, gang violence and internal displacement - rather than on morbidity and mortality impacts of inter-state conflicts or refugees. 

Line 75: Rather than 'victims of internal conflict in Columbia' does this refer rather to those who are internally displaced as a result of the conflict? Again unclear terminology.

Line 60: It is important to highlight here that globally many people forced to migrate due to conflict or violence are also increasingly unable to access protection mechanisms or 'safe havens', due to both their legal status in receiving countries (e.g. Syrian refugees in Lebanon living in situations of informality - as Lebanon has not signed up to the Refugee Convention).

Line 96: Motivation section should include 'access to healthcare for IDPs' as this has been mentioned in the previous paragraphs but is then not mentioned again.

Paragraph from Line 97: again this review has to be adjusted as it focuses more generally on conflict rather than specifically on IDPs or protracted conflict.

Line 101:  'Limited access to health services' rather than 'access to limited health services'? 

Line 109: this sentence:' disruption created by violent conflict affects socioeconomic conditions...' needs a citation. 

Line 111: 'Skepticism' from whom? And what evidence is there? This would need evidence or citation otherwise it should be removed.

Line 113: I think again comparison with other protracted conflicts is necessary here.  In fact, IDPs in protracted conflicts are more often dispersed throughout the country rather than in 'IDP camps' as is implied here e.g. even in Syria the IDPs are highly dispersed.  

Line 134: IDP usually refers to internally displaced persons but in this case it would seem that authors may be referring to internally displaced population instead.  Best to either write in full here or adjust

Line 136: 'They live in poverty' - please provide references and also explain more fully: is this because the social welfare system is not sufficient?

Line: 139: are you linking the displacement with lower socioeconomic conditions - or the protracted conflict?  Again further clarity here would be helpful.

Line 140: The references need to be double checked here - they should be [15] and [16] I believe which refer to the Syrian conflict and NCDs and internal displacement?  Also this conclusion is not clear to me.  Numerous studies have shown no systematic association between migration and importation of communicable diseases.  It is however true that living conditions and conditions of travel may expose those who have been displaced to infectious diseases, but this generally does not result in transmission to host population.  Are you referring to lack of access to health system/vaccinations in situations of conflict as being a risk factor for communicable diseases? This statement (line 140 to 144) needs to either be qualified or further explained.

Line 148: Suggest some more analysis of the limitations of data collection from IDPs here: i.e. not only limitations of the terminology used in the census and lack of anonymity in the census data; but also the fact that many IDPs may not have a permanent place of residence or may move multiple times, and therefore not be included in the census.

Line 165: unsure what 'mortality shock or bust for exogenous reasons' means

Methodology section: It would be valuable to have a clearer explanation of how the subpopulations were separated from the rest of the data.  I was unclear in the data sets how people who had experienced violence/were displaced had been identified, and whether the same terminology was used across the different datasets - this should be explained/outlined. In addition, I would like to have details of any ethical considerations or ethics clearance that took place with regard to accessing and using the data.

After the methodology section there needs to be a discussion section before the conclusions section.  From the data presented I am unclear what conclusions can be drawn about internal displacement/conflict impacts on morbidity and mortality.  It seems that there is on difference in morbidity demonstrated between populations, but there is insufficient discussion of the findings.

In the conclusions section:  I would like to see some further analysis of why those who have been displaced by armed conflict may how worse mortality outcomes for non-communicable diseases i.e. please link back to the topics raised in the introduction around access to healthcare and lack of continuity of care etc.

Thank you. I look forward to receiving the revised article.
